# Structural basis of diverse membrane target recognitions by ankyrins

Chao Wang[1†], Zhiyi Wei[1,2,3†], Keyu Chen[1], Fei Ye[1], Cong Yu[1,3], Vann Bennett[4,5,6], Mingjie Zhang[1,2]*

[1]Division of Life Science, State Key Laboratory of Molecular Neuroscience, Hong Kong University of Science and Technology, Hong Kong, Hong Kong; [2]Center of Systems Biology and Human Health, Hong Kong University of Science and Technology, Hong Kong, Hong Kong; [3]Department of Biology, South University of Science and Technology of China, Shenzhen, China; [4]Department of Biochemistry, Howard Hughes Medical Institute, Duke University Medical Center, Durham, United States; [5]Department of Cell Biology, Duke University Medical Center, Durham, United States; [6]Department of Neurobiology, Duke University Medical Center, Durham, United States

**Abstract** Ankyrin adaptors together with their spectrin partners coordinate diverse ion channels and cell adhesion molecules within plasma membrane domains and thereby promote physiological activities including fast signaling in the heart and nervous system. Ankyrins specifically bind to numerous membrane targets through their 24 ankyrin repeats (ANK repeats), although the mechanism for the facile and independent evolution of these interactions has not been resolved. Here we report the structures of ANK repeats in complex with an inhibitory segment from the C-terminal regulatory domain and with a sodium channel Nav1.2 peptide, respectively, showing that the extended, extremely conserved inner groove spanning the entire ANK repeat solenoid contains multiple target binding sites capable of accommodating target proteins with very diverse sequences via combinatorial usage of these sites. These structures establish a framework for understanding the evolution of ankyrins' membrane targets, with implications for other proteins containing extended ANK repeat domains.

*For correspondence: mzhang@ust.hk

†These authors contributed equally to this work

Competing interests: The authors declare that no competing interests exist.

## Introduction

Ankyrins are a family of scaffold proteins which play essential roles in connecting numerous ion channels, cell adhesion molecules, and receptors to the spectrin-based cytoskeleton beneath membranes and thereby provide mechanical support for plasma membranes and control the activities of excitable tissues including neurons and muscles (*Bennett and Chen, 2001*; *Bennett and Healy, 2009*). There are three members in the human ankyrin family: ankyrin-R/B/G (AnkR/B/G) encoded by *ANK1/2/3*, respectively. They all consist of a highly similar N-terminal membrane binding domain composed of 24 ankyrin (ANK) repeats, a spectrin-binding domain comprised of two ZU5 domains, and a UPA domain followed by a death domain (DD) and a variable C-terminal regulatory domain (*Bennett and Lorenzo, 2013*) (*Figure 1A*). Although sharing similar domain organization, the three ankyrins have distinct and non-overlapping functions in specific membrane domains coordinated by ankyrin-spectrin networks (*Mohler et al., 2002*; *Abdi et al., 2006*; *He et al., 2013*). As ankyrins are adaptor proteins linking membrane proteins to the underlying cytoskeleton, ankyrin dysfunction is closely related to serious human diseases. For example, loss-of-function mutations can cause hemolytic anemia (*Gallagher, 2005*), various cardiac diseases including several cardiac arrhythmia syndromes and sinus node dysfunction (*Mohler et al., 2003, 2007*; *Le Scouarnec et al., 2008*; *Hashemi et al., 2009*), bipolar disorder (*Ferreira et al., 2008*; *Dedman et al., 2012*; *Rueckert et al., 2013*), and autism spectrum disorder (*Iqbal et al., 2013*; *Shi et al., 2013*).

**eLife digest** Proteins are made up of smaller building blocks called amino acids that are linked to form long chains that then fold into specific shapes. Each protein gets its unique identity from the number and order of the amino acids that it contains, but different proteins can contain similar arrangements of amino acids. These similar sequences, known as motifs, are usually short and typically mark the sites within proteins that bind to other molecules or proteins. A single protein can contain many motifs, including multiple repeats of the same motif.

One common motif is called the ankyrin (or ANK) repeat, which is found in 100s of proteins in different species, including bacteria and humans. Ankyrin proteins perform a range of important functions, such as connecting proteins in the cell surface membrane to a scaffold-like structure underneath the membrane.

Proteins containing ankyrin repeats are known to interact with a diverse range of other proteins (or targets) that are different in size and shape. The 24 repeats found in human ankyrin proteins appear to have essentially remained unchanged for the last 500 million years. As such, it remains unclear how the conserved ankyrin repeats can bind to such a wide variety of protein targets.

Now, Wang, Wei et al. have uncovered the three-dimensional structure of ankyrin repeats from a human ankyrin protein while it was bound either to a regulatory fragment from another ankyrin protein or to a region of a target protein (which transports sodium ions in and out of cells). The ankyrin repeats were shown to form an extended 'left-handed helix': a structure that has also been seen in other proteins with different repeating motifs. Wang, Wei et al. found that the ankyrin protein fragment bound to the inner surface of the part of the helix formed by the first 14 ankyrin repeats. The target protein region also bound to the helix's inner surface. Wang, Wei et al. show that this surface contains many binding sites that can be used, in different combinations, to allow ankyrins to interact with diverse proteins.

Other proteins with long sequences of repeats are widespread in nature, but uncovering the structures of these proteins is technically challenging. Wang, Wei et al.'s findings might reveal new insights into the functions of many of such proteins in a wide range of living species. Furthermore, the new structures could help explain why specific mutations in the genes that encode ankyrins (or their binding targets) can cause various diseases in humans—including heart diseases and psychiatric disorders.

The wide-ranging physiological functions of ankyrins are the result of ankyrin's remarkable capacity for binding to and anchoring numerous membrane targets, via their N-terminal 24 ANK repeats, to specific membrane micro-domains in coordination with spectrin-based cytoskeletal structures (*Bennett and Chen, 2001*). One good example is the formation and maintenance of axon initial segments (AIS) in neurons. Interaction between AnkG and L1 cell adhesion molecules (e.g., the 186 kDa neurofascin, referred to as Nfasc in this study) is required for the formation and stability of the AIS (*Hedstrom et al., 2007*; *Zonta et al., 2011*). AnkG is in turn responsible for clustering voltage-gated sodium channels at the AIS, which underlies the mechanistic basis of action potential generation and propagation (*Garrido et al., 2003*; *Kole et al., 2008*) (see review by *Rasband, 2010*). Anchoring of L1 cell adhesion molecules also directs inhibitory GABAergic synapse innervation at the AIS of excitatory Purkinje neurons (*Ango et al., 2004*), a critical step for balanced neuronal circuit formation. Depletion of AnkG both in cultured hippocampal neurons and in mice causes axons to lose axonal properties and acquire the molecular characteristics of dendrites, showing that AnkG is required for the maintenance of axonal polarity (*Hedstrom et al., 2008*; *Sobotzik et al., 2009*).

Other examples of ankyrin function in organizing membrane signaling networks include, but are not limit to, AnkB/G-mediated coordination of voltage-gated sodium channels, Na/K ATPase, Na/Ca exchanger, and inositol 1,4,5-triphosphate receptors in cardiomyocytes (*Mohler et al., 2004*, *2005*; *Hund et al., 2008*; *Lowe et al., 2008*) and the dystrophin/dystroglycan complex in skeletal muscles (*Ayalon et al., 2008*, *2011*). Finally, as originally discovered by *Bennett and Stenbuck (1979a*, *1979b)*, AnkR is well known to be essential for preserving erythrocyte membrane integrity. Although the critical functions of ankyrins in the specialized membrane domains have been recognized for decades, the underlying mechanistic basis governing ankyrin's coordination with such broad spectrum of membrane

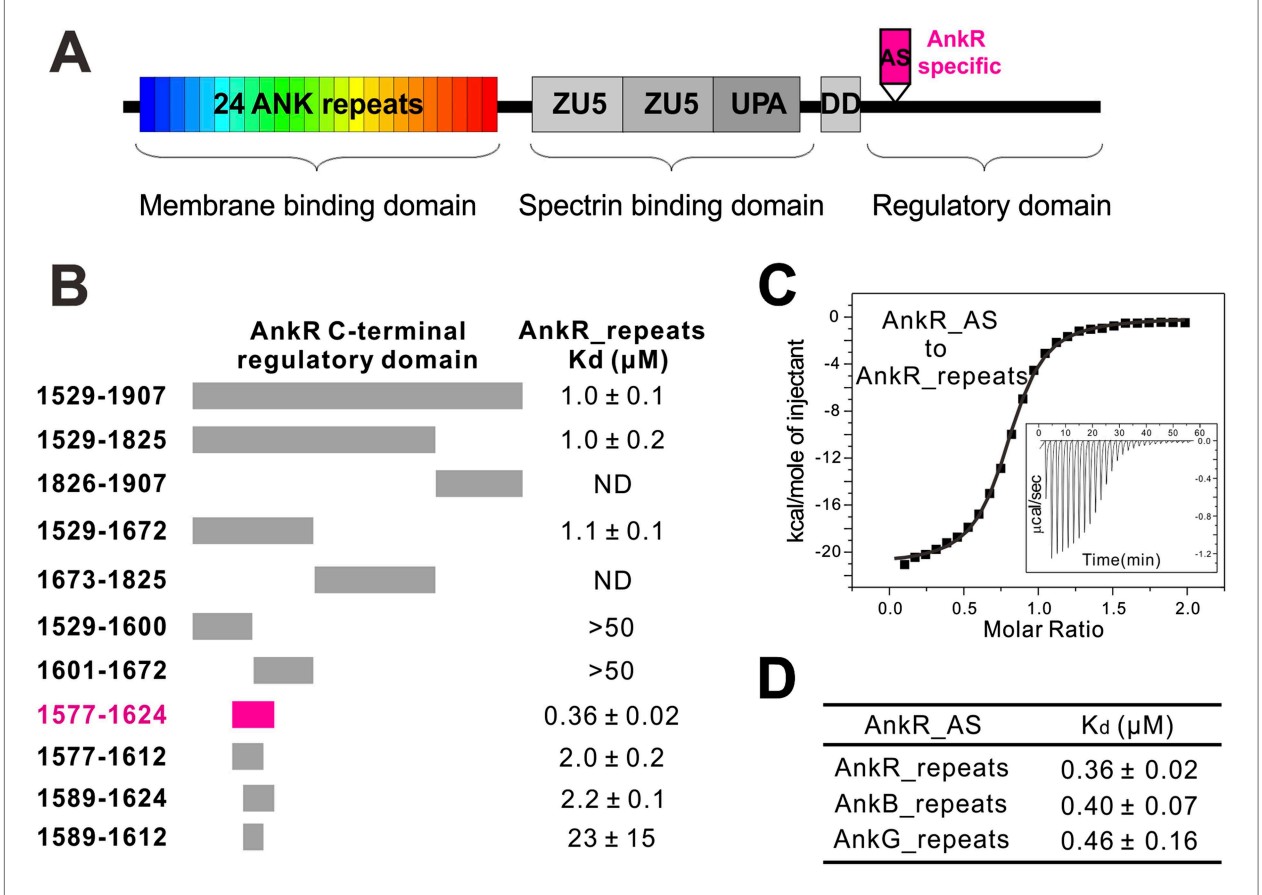

**Figure 1**. Identification of a 48-residue auto-inhibitory segment that binds to ANK repeats. (**A**) Schematic diagrams showing the domain organization of ankyrins. The AnkR-specific auto-inhibitory segment (AS) is indicated within the C-terminal regulatory domain. The same color codes (24 ANK repeats in rainbow and the AnkR_AS in magenta) are used throughout the paper unless otherwise stated. (**B**) ITC-based mapping of the minimal AnkR_repeats binding region in the C-terminal regulatory domain. The minimal and complete AS identified is highlighted in magenta. 'ND' denotes that these constructs had no detectable binding to ANK repeats. (**C**) ITC-derived binding curve of AnkR_AS titrated to AnkR_repeats. (**D**) The binding affinities between AS and ANK repeats of the three ankyrin isoforms.

targets remains essentially unknown, largely due to challenges in characterizing the biochemical and structural properties of the elongated ANK repeats. Additionally, it is noted that the ANK repeats of ankyrins have been extremely conserved, whereas the membrane targets have continued to expand throughout evolution, presumably due to functional requirements for membrane microdomain-mediated fast signaling events in higher eukaryotes including mammals.

In this study, we performed detailed biochemical characterizations of ANK repeats of ankyrins and their interactions with various binding partners. We solved the crystal structures of ANK repeats in complex with an auto-inhibitory segment from AnkR C-terminal domain and with a peptide from Nav1.2, respectively. The 24 ANK repeats of ankyrins form a superhelical solenoid with an extremely conserved elongated inner groove, which contains multiple quasi-independent target binding sites. We further show that ankyrins can accommodate different membrane targets with diverse sequences by combinatorial usage of these binding sites. The ankyrin-Nav1.2 complex structure also provides a mechanistic explanation for the mutation found in Nav channels that causes cardiac disease in humans. Collectively, our findings provide a first glimpse into the mechanistic basis governing membrane target recognition by the highly conserved ANK repeats in ankyrins and establish a structural framework for future investigation of ankyrin's involvement in physiological functions and pathological conditions in diverse tissues. Our results also provide a molecular mechanism for the rapid expansion of ankyrin partners in vertebrate evolution. These insights also will be valuable for understanding the recognition mechanisms of other long ANK repeat proteins as well as many other long repeat-containing proteins in living organisms in general.

## Results

### An auto-inhibitory segment from the C-terminal domain of AnkR specifically binds to ANK repeats of ankyrins

To elucidate the mechanisms governing ANK repeat-mediated binding of ankyrins to diverse membrane targets, we attempted to determine the atomic structures of ANK repeats alone or in complex with their targets. However, extensive trials of crystallizing ANK repeat domains of AnkR/B/G were not successful, presumably because of the highly dynamic nature of the extended ANK repeat solenoid (*Howard and Bechstedt, 2004*; *Lee et al., 2006*). Anticipating that ANK repeats binders may rigidify the conformation of ANK repeats, we turned our attention to the ANK repeat/target complexes. The C-terminal regulatory domains have been reported to bind to ANK repeats intra-molecularly and modulate the target binding properties of ankyrins (*Davis et al., 1992*; *Abdi et al., 2006*). We measured the interaction of AnkR_repeats with its entire C-terminal regulatory domain (residues 1529–1907) using highly purified recombinant proteins, and found that they interact with each other with a $K_d$ of around 1 μM (*Figure 1B*). It is expected that the intra-molecular association between ANK repeats and its C-terminal tail of AnkR is very stable, and thus the full-length AnkR likely adopts an auto-inhibited conformation and ANK repeats-mediated binding to membrane targets requires release of the auto-inhibited conformation of AnkR.

Using isothermal titration calorimetry (ITC)-based quantitative binding assays, we identified a 48-residue auto-inhibitory segment (residues 1577–1624, referred to as 'AS') as the complete ANK repeat-binding region (*Figure 1B,C*). Further truncation at either end of this 48-residue AS fragment significantly decreased its binding to AnkR_repeats (*Figure 1B*). The corresponding sequence does not exist in AnkB or AnkG, indicating the AS is specific to AnkR (*Figure 1A*). AnkR_AS was found to bind to AnkR/B/G ANK repeats with comparable affinities (*Figure 1D*), as expected since AnkR/B/G share extremely conserved ANK repeat sequences (*Figure 2B* and see below). Thus, we tried the complexes of AnkR_AS with ANK repeats of all three isoforms to increase the chances of obtaining suitable crystals. Although crystals of various complexes were obtained, they all diffracted very poorly. After extensive trials of screening and optimization, we succeeded in obtaining good-diffraction crystals of AnkR_AS fused at its C-terminus with the AnkB_repeats and solved the structure of the fusion protein at 3.5 Å resolution (*Figure 2C* and *Table 1*). The NMR spectra of the $^{13}CH_3$-Met selectively labeled fusion protein and the ANK repeats/AS complex produced by cleavage of the fusion protein at the fusion site are essentially identical (*Figure 2—figure supplement 1*), indicating that the fusion strategy used here facilitates crystallization but does not alter the structure of the ANK repeats/AS complex. There are three Met residues in AS (Met1601, Met1604, and Met1607) and all three Met residues are in the binding interface between ANK repeats and AS (*Figure 2—figure supplement 2A*).

### Overall structure of the AnkB_repeats/AnkR_AS complex

Except for a few connecting loops and termini of the chains, the rest of the ANK repeats and AS are properly defined (*Figure 2C* and *Figure 2—figure supplement 2*). The 24 ANK repeats form a left-handed helical solenoid with each repeat rotating anti-clockwise by ~16° (*Figure 2C*). Except for the capping helices in the first and last repeats (i.e., αA of R1 and αB of R24), each repeat has the typical ANK repeat sequence pattern and forms a helix-turn-helix conformation (*Figure 2A,C*). A well-defined finger-like hairpin loop (finger loop) connects two consecutive repeats. The inner αA helices and the finger loops of the 24 repeats line together to form an elongated concave inner groove, and the αB helices of the repeats form the solvent-exposed convex outer surface. The ANK repeats super-helix has outer and inner diameters of approximately 60 Å and 45 Å, respectively, and a total height of ~150 Å (*Figure 2C*). The size of the ANK repeats revealed here is consistent with the previous measurement by atomic force microscopy (*Lee et al., 2006*). The C-terminal half of the ANK repeats structure aligns well with the apo-form structure of the last 12 ANK repeats of AnkR with an overall r.m.s.d. of 1.6 Å (*Michaely et al., 2002*).

We analyzed the amino acid residues at each position of vertebrate AnkR/B/G ANK repeats and found that conservation is above 80% at most of the positions (*Figure 2B* and *Figure 2—figure supplement 3*). Further analysis reveals that residues forming the target binding concave inner groove (i.e., residues of the finger loops and αA helices of the 24 repeats) are essentially identical among vertebrate AnkR/B/G (*Figure 2B* and *Figure 2—figure supplement 3*), indicating that both the structure and the target binding properties of their ANK repeats are likely to be the same (also see *Figure 1D*).

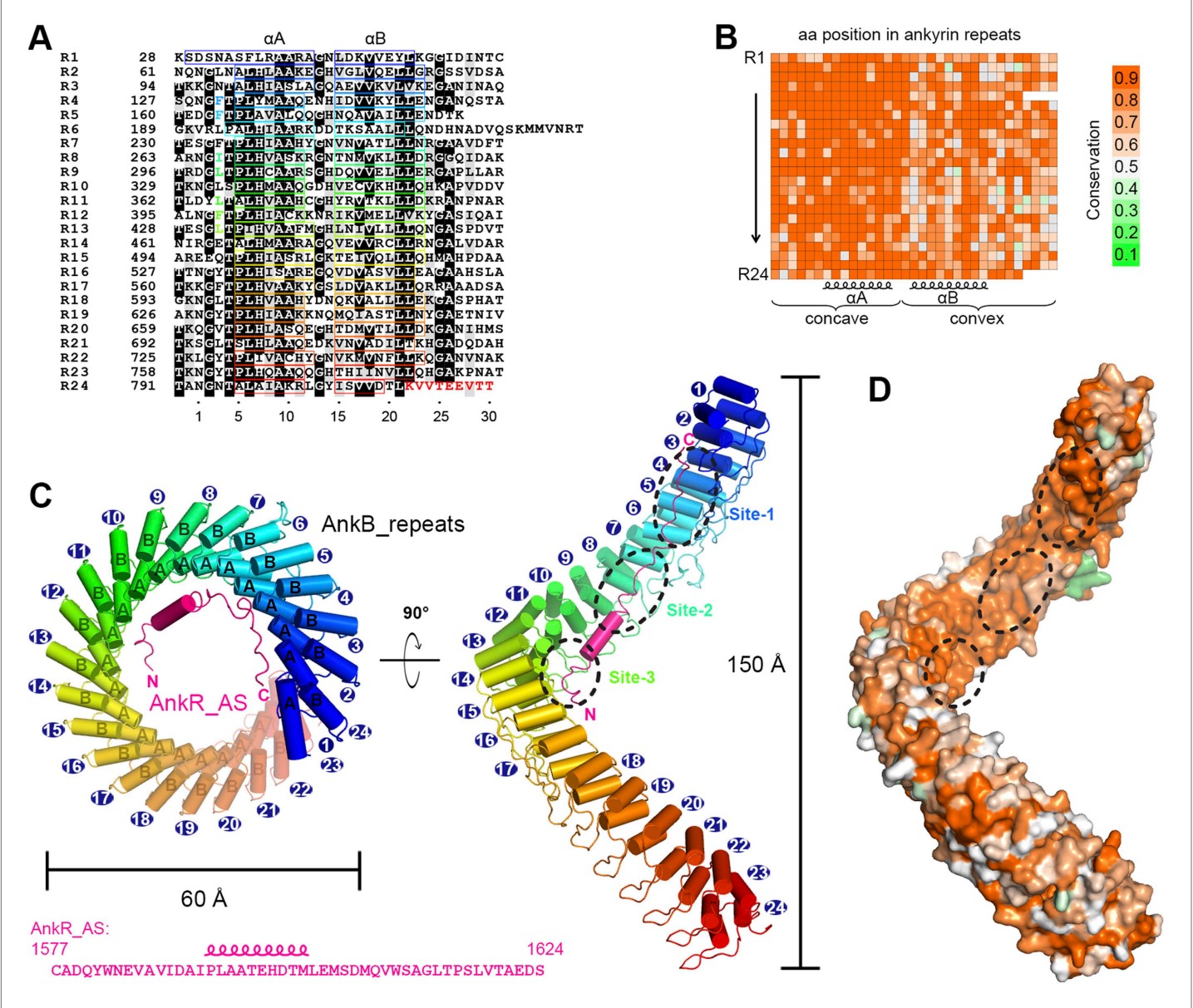

**Figure 2**. Vertebrate ANK repeats of ankyrins share the same architecture and target binding properties. (**A**) Sequence alignment of the 24 ANK repeats of human AnkB. Similar and identical residues are labeled gray and black, respectively. The helix formation residues are boxed with corresponding colors. The hydrophobic residues selected for mutation studies described in *Figure 3* and onwards are labeled with corresponding colors. The last nine amino acids labeled in red from R24 are used as the C-terminal capping sequence for designed truncation mutants of various lengths of ANK repeats used in this study. (**B**) Sequence conservation map of the 24 ANK repeats of vertebrate ankyrins. The conservation score for each residue is calculated based on the sequences of vertebrate ankyrins aligned in *Figure 2—figure supplement 3* through the Scorecons server (http://www.ebi.ac.uk/thornton-srv/databases/cgi-bin/valdar/scorecons_server.pl). The position of each residue is the same as that shown in panel **A**. (**C**) Overall structure of the ANK repeats/AS complex viewed from the top (left) and side (right). The three AS-binding surfaces on ANK repeats are circled with black dashed ovals. The sequences of AnkR_AS are listed below. (**D**) Surface conservation map of ANK repeats viewed from the side. The conservation map is derived from the ankyrins from worm to human as shown in *Figure 2—figure supplement 3* with the same color coding scheme as in panel (**B**).

The following figure supplements are available for figure 2:

**Figure supplement 1**. The fusion of AnkR_AS to the N-terminus AnkB_repeats does not alter the conformation of the ANK repeats/AS complex.

**Figure supplement 2**. Crystallographic characterization of the ANK repeats/AS structure.

**Figure supplement 3**. Amino acid sequence alignment of ANK repeats of ankyrins.

**Table 1.** Statistics of data collection and model refinement

| | Native ANK repeats/AS | SeMet-ANK repeats/AS | R1-9/Nav1.2_ABD-C |
|---|---|---|---|
| Data collection | | | |
| Space group | $R32$ | $R32$ | $P4_222$ |
| Cell dimensions | | | |
| $a, b, c$ (Å) | 179.9, 179.9, 304.5 | 179.7, 179.7, 304.9 | 102.3, 102.3, 106.0 |
| $\alpha, \beta, \gamma$ (°) | 90, 90, 120 | 90, 90, 120 | 90, 90, 90 |
| Resolution range (Å) | 50–4.0 (4.07–4.0) | 50–3.5 (3.56–3.5) | 50–2.5 (2.54–2.5) |
| $R_{merge}$ (%)* | 8.7 (45.8) | 12.1 (78.3) | 7.7 (74.8) |
| $I/\sigma I$ | 17.1 (3.4) | 22.5 (2.2) | 29.8 (3.5) |
| Completeness (%) | 98.9 (99.3) | 96.0 (97.2) | 99.4 (100) |
| Redundancy | 4.3 (4.4) | 10.2 (9.0) | 9.5 (9.7) |
| Structure refinement | | | |
| Resolution (Å) | | 50–3.5 (3.62–3.5) | 50–2.5 (2.64–2.5) |
| $R_{cryst}/R_{free}$ (%)† | | 22.0 (35.0)/25.3 (36.6) | 18.8 (22.7)/23.8 (24.5) |
| r.m.s.d. bonds (Å)/angles (°) | | 0.013/1.5 | 0.015/1.5 |
| Average B factor | | 113.5 | 63.5 |
| No. of atoms | | | |
| Protein atoms | | 6260 | 2243 |
| Water molecules | | 0 | 74 |
| Other molecules | | 45 | 57 |
| Ramachandran plot‡ | | | |
| Favored regions (%) | | 94.7 | 97.7 |
| Allowed regions (%) | | 5.2 | 2.3 |
| Outliers (%) | | 0.1 | 0.0 |

*$R_{merge} = \Sigma|I_i - I_m|/\Sigma I_i$, where $I_i$ is the intensity of the measured reflection and $I_m$ is the mean intensity of all symmetry related reflections.
†$R_{cryst} = \Sigma||F_{obs}| - |F_{calc}||/\Sigma|F_{obs}|$, where $F_{obs}$ and $F_{calc}$ are observed and calculated structure factors, respectively. $R_{free} = \Sigma_T||F_{obs}| - |F_{calc}||/\Sigma_T|F_{obs}|$, where T is a test data set of about 5% of the total reflections randomly chosen and set aside prior to refinement.
‡Defined by MolProbity.
Numbers in parentheses represent the value for the highest resolution shell.

Additionally, the residues in the entire inner groove of the ANK repeats superhelix are highly conserved for all ankyrins throughout evolution (from worm to human) (*Figure 2D* and *Video 1*), suggesting that the functions of ANK repeats in different species of ankyrins are highly conserved during evolution and that the inner groove of ANK repeats is the general binding site for membrane-associated targets of ankyrins. Consistent with this prediction, binding of AS to AnkG_repeats prevents voltage-gated sodium channel Nav1.2 and Nfasc from binding to AnkG (*Figure 3—figure supplement 1*). Therefore, we hypothesized that the ANK repeats/AS structure presented here serves as a general framework for understanding how ankyrins engage their membrane targets, and tested this hypothesis using mutations designed and tested as described below.

Before binding to ANK repeats, AS adopts a random coil structure as indicated by its NMR spectrum (data not shown). In the complex, AS adopts a highly extended structure binding to part of the inner groove formed by the N-terminal 14 ANK repeats (R1–14) with its chain orientation anti-parallel to that of ANK repeats (*Figure 2A,C*). A 10-residue segment of AS (residues 1592–1601) forms an α helix when bound to ANK repeats (*Figure 2C*). The residues connecting AS and ANK repeats (10 residues in total, 'GSLVPRGSGS') are flexible, indicating that the fusion of the two chains together does not introduce obvious conformational restraints to the complex.

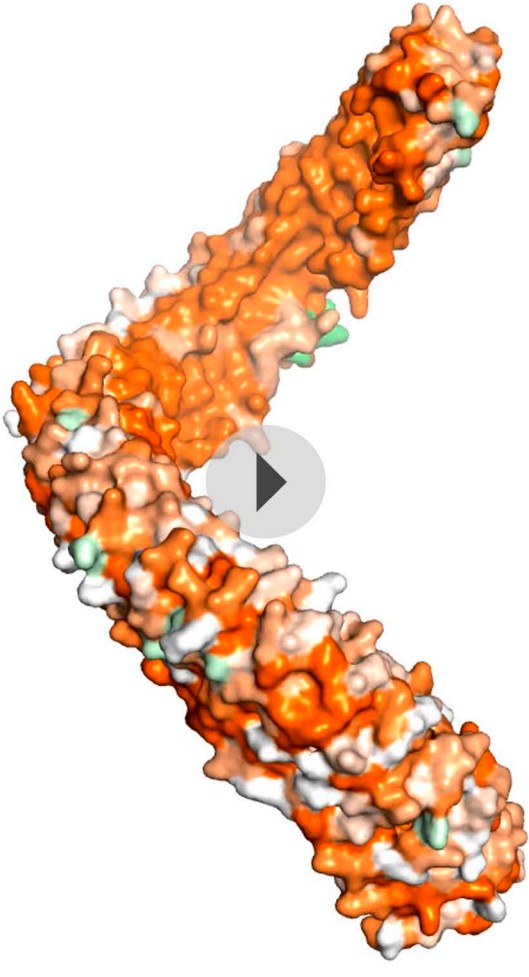

**Video 1**. Surface conservation of 24 ANK repeats. This video shows the concave groove is highly conserved across various species from human to worm.

The binding of AS to ANK repeats can be divided somewhat arbitrarily into three sites (sites 1, 2, and 3) formed by the repeats 2–6, 7–10, and 11–14, respectively (*Figure 2C* and *Figure 3A–C*). Nonetheless, this division is supported by several lines of evidence. Structurally, there is a fairly clear boundary between each of the two binding sites in the ANK repeats/AS complex structure, whereas the interactions within each site are rather concentrated (*Figure 3*). The most direct evidence is from the interaction between ANK repeats and Nav1.2 (see below). In the case of Nav1.2 binding, R1–6 of ANK repeats binds to the C-terminal half of the Nav1.2_ABD (ankyrin binding domain) and R11–14 binds to the N-terminal half of Nav1.2_ABD. R7–10 is not involved in the Nav1.2 binding. Thus, one can naturally divide ANK repeats R1–14 into three parts. Such division is further supported by the accepted concept that four to five ANK repeats can form a folded structural unit. In our case, sites 2 and 3 contain four repeats each, and site 1 contains five repeats if we do not count the repeat 1 which serves as a capping repeat.

The interactions in site 1 are primarily charge–charge and hydrogen bonding in nature, although hydrophobic contacts also contribute to the binding (*Figure 3A*). The interactions in site 2 are mediated both by hydrophobic and hydrogen bonding interactions, while interactions in site 3 are mainly hydrophobic (*Figure 3B,C*). The structure of the ANK repeats/AS complex is consistent with the idea that ANK repeats bind to relatively short and unstructured peptide segments in ankyrins' membrane targets (*Bennett and Healy, 2009*; *Bennett and Lorenzo, 2013*).

## Ankyrins bind to Nav1.2 and Nfasc through combinatorial usage of multiple binding sites

We next examined the interactions of AnkG_repeats with Nav1.2 and Nfasc using the structure of the ANK repeats/AS complex to design mutations specifically affecting each predicted site. The $K_d$ of the binding of AnkG_repeats to the Nav1.2_ABD (residues 1035–1129, comprising the majority of the cytoplasmic loop connecting transmembrane helices II and III, see below for details) and to the Nfasc_ABD (a 28-residue fragment in the cytoplasmic tail; *Figure 3—figure supplement 2* and see *Garver et al., 1997*) is 0.17 and 0.21 μM, respectively (*Figure 3E*, upper panels). To probe the binding sites of Nav1.2 and Nfasc on AnkG, we constructed AnkG_repeat mutants with the corresponding hydrophobic residues in binding site 1 (Phe131 and Phe164 in R4 and R5, termed 'FF'), site 2 (Ile267 and Leu300 in R8 and R9; 'IL'), and site 3 (Leu366, Phe399, and Leu432 in R11, R12, and R13; 'LFL') substituted with Gln (*Figure 3D*), and examined their binding to the two targets. The mutations in site 1 significantly decreased ANK repeat binding to Nav1.2, but had no impact on Nfasc binding. Conversely, the mutations in site 2 had minimal impact on Nav1.2 binding, but significantly weakened Nfasc binding. The mutations in site 3 weakened ANK repeat binding to both targets (*Figure 3F*, *Figure 3—figure supplement 3* and *Figure 3—figure supplement 4*). The above results indicate that the two targets bind to ANK repeats with distinct modes, with Nav1.2 binding to sites 1 and 3 and Nfasc binding to sites 2 and 3. This conclusion is further supported by the binding of the two targets to various AnkG_repeat truncation mutants (*Figure 3F*, *Figure 3—figure supplement 3* and *Figure 3—figure supplement 4*).

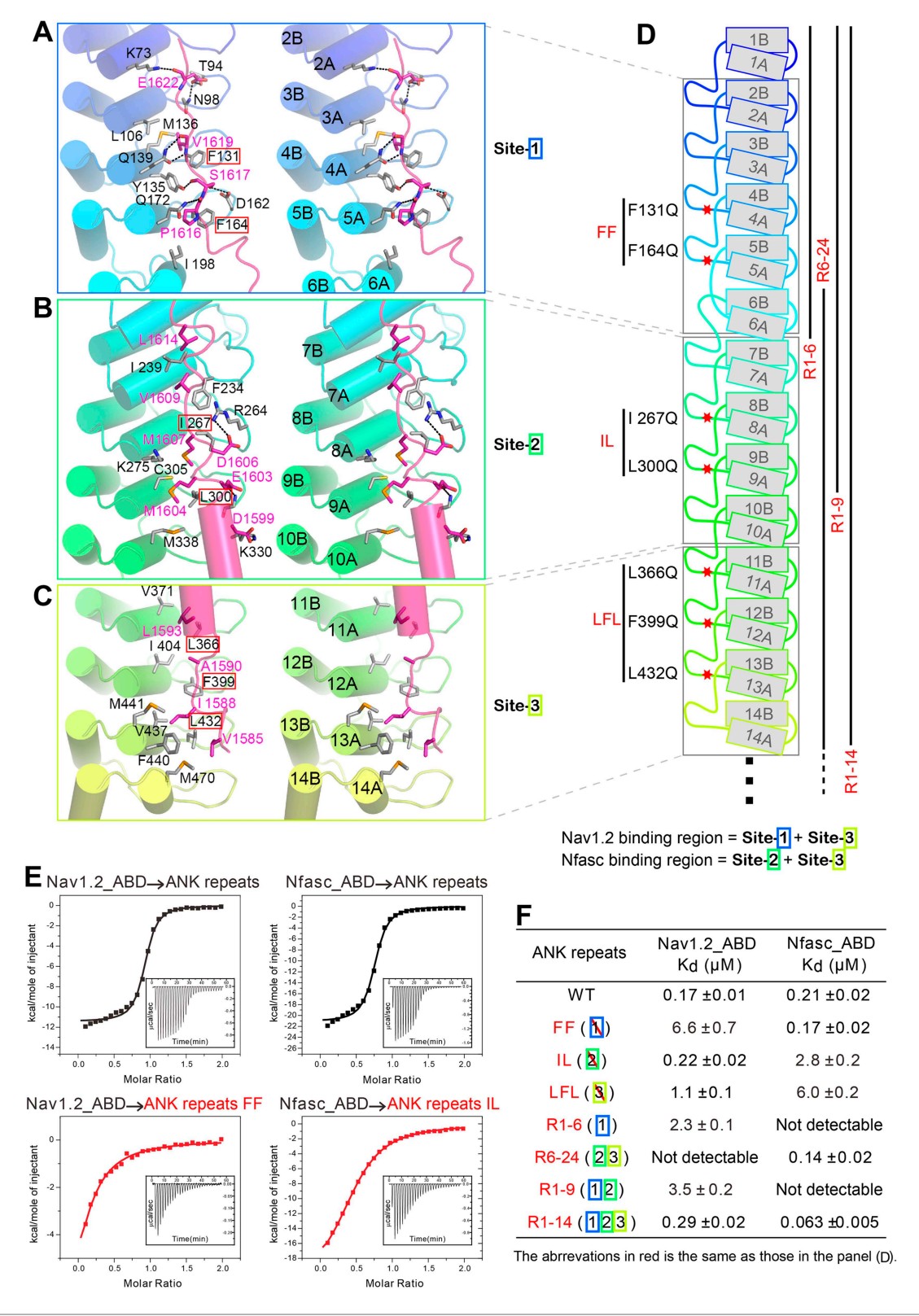

**Figure 3**. Structural and biochemical characterizations of target binding properties of ANK repeats. (**A–C**) Stereo views showing the detailed ANK repeats/AS interfaces of the three binding sites shown in **Figure 1E**. Hydrogen bonds and salt bridges are indicated by dashed lines. (**D**) Cartoon diagram of the first 14 repeats of the 24 ANK repeats. Different truncations used for the biochemical analyses are indicated below. Mutations of hydrophobic

*Figure 3. Continued on next page*

*Figure 3. Continued*

residues in the three AS binding sites are labeled. Red stars indicate the locations of the mutation sites. (**E**) Example ITC curves showing the bindings of Nav1.2_ABD or Nfasc_ABD to the wild-type or mutant ANK repeats. (**F**) The dissociation constants of the binding reactions of various mutants of ANK repeats to Nav1.2 and Nfasc derived from the ITC-based assays.

The following figure supplements are available for figure 3:

**Figure supplement 1**. Analytical gel filtration analyses showing that binding of AS to AnkG_repeats prevents Nav1.2 and Nfasc ABDs from binding to AnkG_repeats.

**Figure supplement 2**. ITC-based analyses of the AnkG_repeats/Nfasc_ABD interaction.

**Figure supplement 3**. The ITC curves of the bindings of various ANK repeats to Nav1.2_ABD.

**Figure supplement 4**. The ITC curves of the bindings of various ANK repeats to Nfasc_ABD.

We have also assayed the impact of the mutations of the three sites on the binding of AnkR_AS to ANK repeats. The mutations in sites 1 and 2 led to ~20-fold decrease in AnkR_AS binding, while the site 3 mutation only caused an approximately threefold decrease in AnkR_AS binding (*Figure 4A*). Finally, we tested the binding of another two reported ankyrin targets, the KCNQ2 potassium channel (*Pan et al., 2006*) and the voltage-gated calcium channel Cav1.3 (*Cunha et al., 2011*), to the ANK repeats and its mutants, and found that KCNQ2 mainly binds to sites 1 and 2, and Cav1.3 primarily relies on site 2 of ANK repeats (*Figure 4B,C*). Taken together, the above biochemical analysis plus the structure of the ANK repeats/AS complex reveals that through combinations of multiple binding sites on the extremely conserved and elongated inner groove formed by the 24 ANK repeats, ankyrins can bind to numerous targets with diverse amino acid sequences. It is likely that some ankyrin targets may bind to the groove formed by the rest of the repeats in addition to R1–14.

## An elongated fragment of Nav1.2 binds to ANK repeats

To further delineate the target binding mechanisms of ankyrins, we characterized the interaction between AnkG_repeats and Nav1.2 in detail. Previous studies have reported that the intracellular loop

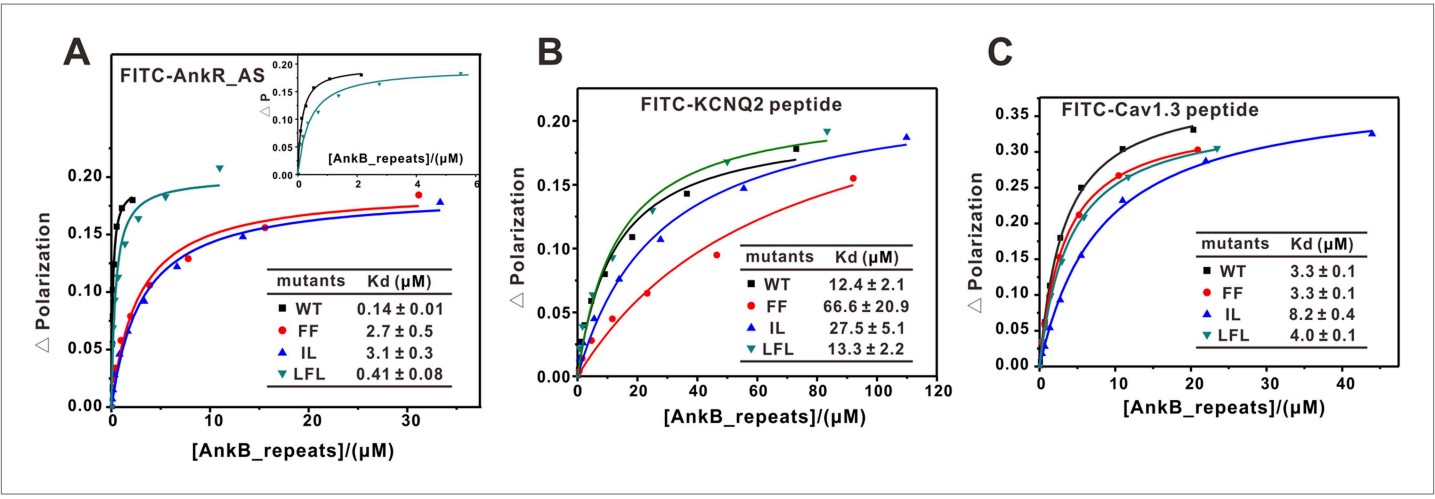

**Figure 4**. Fluorescence polarization-based measurement of the binding affinities of different targets to AnkB_repeats WT and its mutants. (**A**) Fluorescence polarization-based measurement of the binding affinities of AnkR_AS peptide to AnkB_repeats WT and its mutants. The insert shows the expanded view of the binding curves of the AnkR_AS peptides to WT and LFL of AnkB_repeats. The binding affinity between AnkR_AS and AnkB_repeats WT measured through this experiment is slightly different from the ITC assay (0.14 µM vs 0.40 µM). This may be because of the different measuring system, but the overall affinity range is quite similar. (**B**) Fluorescence polarization-based measurement of the binding affinities of the KCNQ2 peptide to AnkB_repeats WT and its various mutants. (**C**) Fluorescence polarization-based measurement of the binding affinities of the Cav1.3 peptide to AnkB_repeats and its various mutants. The fitted binding affinities are shown within the corresponding figures.

connecting the transmembrane helices II and III (loop 2) is responsible for targeting Nav1.2 to the AIS via directly binding to AnkG, and identified a 27-residue motif within loop 2 ('ABD-C', indicated in *Figure 5A,D*) as the AnkG binding domain (*Garrido et al., 2003*; *Lemaillet et al., 2003*). First, we confirmed that a 95-residue fragment (ABD, residues 1035–1129; *Figure 5D*) is sufficient for binding to AnkG (*Figure 3E*, upper left panel). Surprisingly, we found that the C-terminal part of the ABD (ABD-C, the 27-residue motif identified previously for ANK repeats binding) binds to ANK repeats with an affinity ~15-fold weaker than the entire ABD, indicating that the ABD-C is not sufficient for binding to ANK repeats (*Figure 5B,C*). Consistent with this observation, the N-terminal 68-residue fragment of loop 2 (ABD-N, residues 1035–1102) also binds to ANK repeats, albeit with a relatively weak affinity ($K_d$ of ~8 µM; *Figure 5B,C*). We further showed that the ABD-C fragment binds to repeats 1–6 (R1–6) of ANK repeats, as ABD-C binds to R1–6 and the entire 24 ANK repeats with essentially the same affinities (*Figure 5B,C*). These results also reveal that, like the AnkR_AS, the Nav1.2 peptide segment binds to ANK repeats in an anti-parallel manner. Taken together, the biochemical data shown in *Figure 3E* and *Figure 5* indicate that two distinct fragments of Nav1.2 loop 2, ABD-N and ABD-C, are responsible for binding to ANK repeats. The previously identified ABD-C binds to site 1 and ABD-N binds to site 3 of ANK repeats, and the interactions between the two sites are largely independent from each other energetically.

We noted from the amino acid sequence alignment of the Nav1 members that the sequences of ABD-C (the first half in particular) are much more conserved than those of ABD-N (*Figure 5D*). Further mapping experiments showed that the C-terminal less-conserved 10 residues of ABD-C are not essential for Nav1.2 to bind to ANK repeats (*Figure 5B*, top two rows). Truncations at the either end of Nav1.2 ABD-N weakened its binding to ANK repeats (data not shown), indicating that the entire ABD-N is required for the channel to bind to site 3 of ANK repeats. The diverse ABD-N sequences of Nav1 channels fit with the relatively non-specific hydrophobic-based interactions in site 3 observed in the structure of ANK repeats/AS complex (*Figure 3C*).

## Structure of Nav1.2_ABD-C/AnkB_repeats_R1–9 reveals binding mechanisms

Although with very low amino acid sequence similarity, the Nav1.2_ABD-C (as well as the corresponding sequences from Nav1.5, KCNQ2/3 potassium channels, and β-dystroglycan [*Mohler et al., 2004*; *Pan et al., 2006*; *Ayalon et al., 2008*]) and the site 1 binding region of AnkR_AS share a common pattern with a stretch of hydrophobic residues in the first half followed by a number of negatively charged residues in the second half (*Figure 6C*). Based on the structure of the ANK repeats/AS complex, we predicted that the Nav1.2_ABD-C may also bind to site 1 of AnkG_repeats with a pattern similar to the AS peptide. We verified this prediction by determining the structure of a fusion protein with the first nine ANK repeats of AnkB fused at the C-terminus of Nav1.2_ABD-C at 2.5 Å resolution (*Figure 6A*, *Figure 6—figure supplement 1* and *Table 1*; the ANK repeats/the entire ABD complex crystals diffracted very poorly, presumably because of the flexible nature of the interaction between Nav1.2_ABD-N and site 3 of ANK repeats).

In the complex structure, the extended Nav1.2_ABD-C peptide interacts with the surface of the inner groove formed by the first five ANK repeats (*Figure 6A*). In particular, the hydrophobic residues of Nav1.2_ABD-C and AS occupy very similar positions on the hydrophobic groove formed by residues from ANK repeats R4 and R5, and subtle conformational differences in the finger loops of R4 and R5 can accommodate amino acid sequence differences between the two targets (*Figure 6E*). This similar pattern and subtle accommodation illustrate that ANK repeats in general are incredibly adaptable and versatile as protein binding modules. Unique to Nav1.2, the binding of ABD-C extends all the way to R1 via charge–charge and hydrogen-bond interactions (*Figure 6A,E*). We also compared our ANK repeats complex structure with two recently determined peptide-bound ANK repeats structures, ANKRA2 and RFXANK in complex with HDAC4 and RFX5 peptides, respectively (*Xu et al., 2012*). Although the HDAC4 and RFX5 peptides also bind to ANKRA2 and RFXANK ANK repeats in extended conformations, the key target binding residues are restricted to a small set of hydrophobic residues in the A helices of the five ANK repeats. Accordingly, a consensus sequence motif can be recognized to bind to the ANKRA2 and RFXANK ANK repeats.

## A completely conserved Glu in ABD-C anchors Nav1 to ankyrins

We noted that Glu1112, which is completely conserved in both Na+ and K+ channels and mutation of which in Nav1.5 to Lys is known to cause Brugada syndrome in humans (*Mohler et al., 2004*), occupies

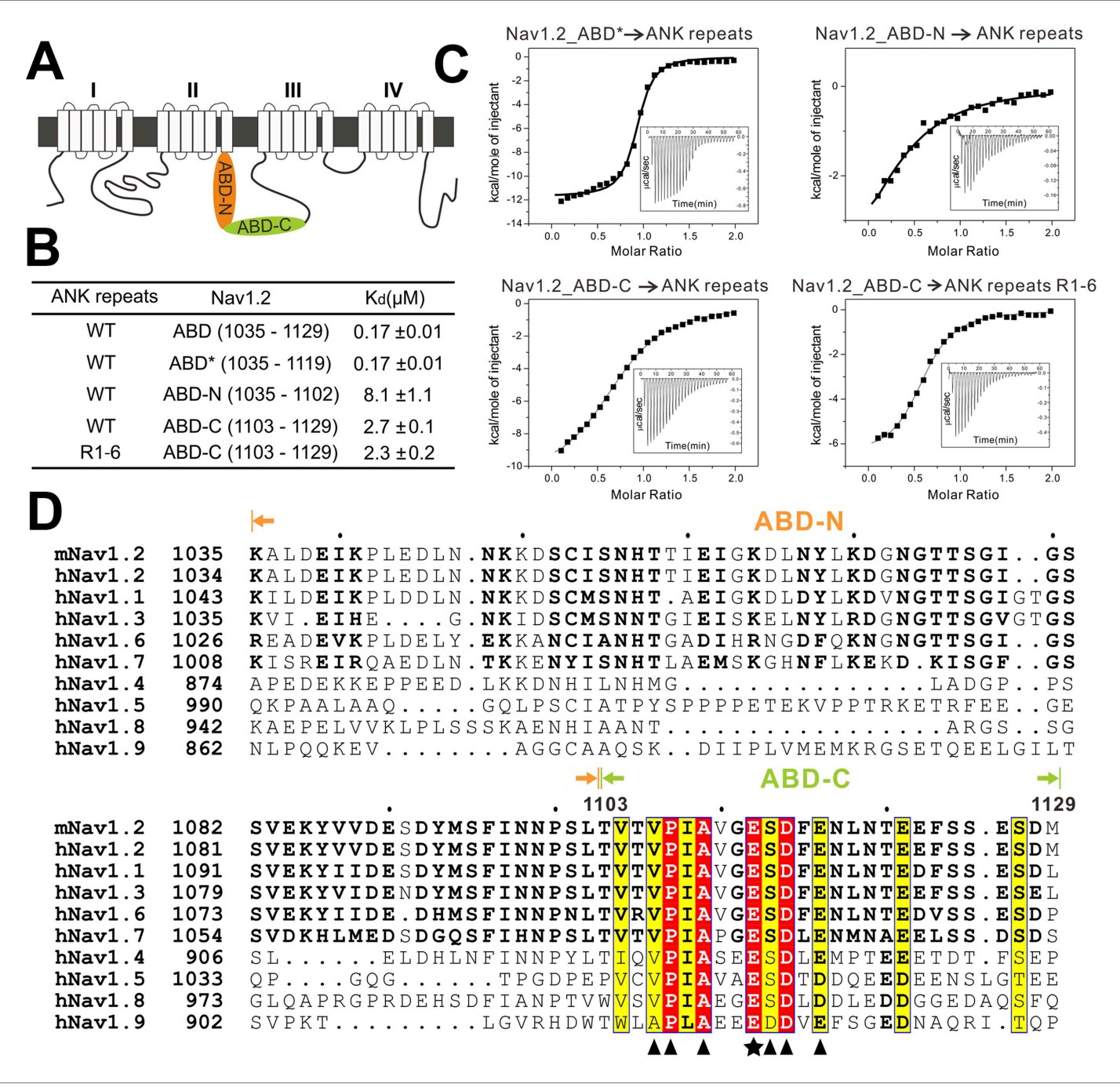

**Figure 5.** Characterization of the interaction between Nav1.2 and AnkG_repeats. (**A**) Schematic diagram showing the domain organization of the Nav1 family ion channels. The ABD is located within loop 2 linking the transmembrane helices II and III and separated into N and C parts according to the data below. (**B**) Table summarizing the ITC-derived affinities of the bindings of various loop 2 fragments to AnkG_repeats. (**C**) ITC curves of the bindings of Nav1.2_ABD* (upper left), ABD-N (upper right), and ABD-C (lower left) to ANK repeats, and Nav1.2_ABD-C binding to ANK repeats R1–6 (lower right), showing that ABD-C binds to site 1 of AnkG_repeats. (**D**) Amino acid sequence alignment of the ankyrin binding domains (ABD) of members of the voltage-gated sodium channel α-subunits (Nav1) family. The mouse Nav1.2 used in this study was aligned with the human family members. Residues that are absolutely conserved and highly conserved are highlighted in red and yellow, respectively. The critical Glu1112 for the binding of Nav1.2 to the ANK repeats is indicated with a star. Other residues participating in the binding with the ANK repeats are indicated by triangles. The residues responsible for binding to site 1 of AnkG_repeats are completely conserved in all members of the Nav1 family, indicating that all sodium channels can bind to ankyrins following the mode revealed in this study.

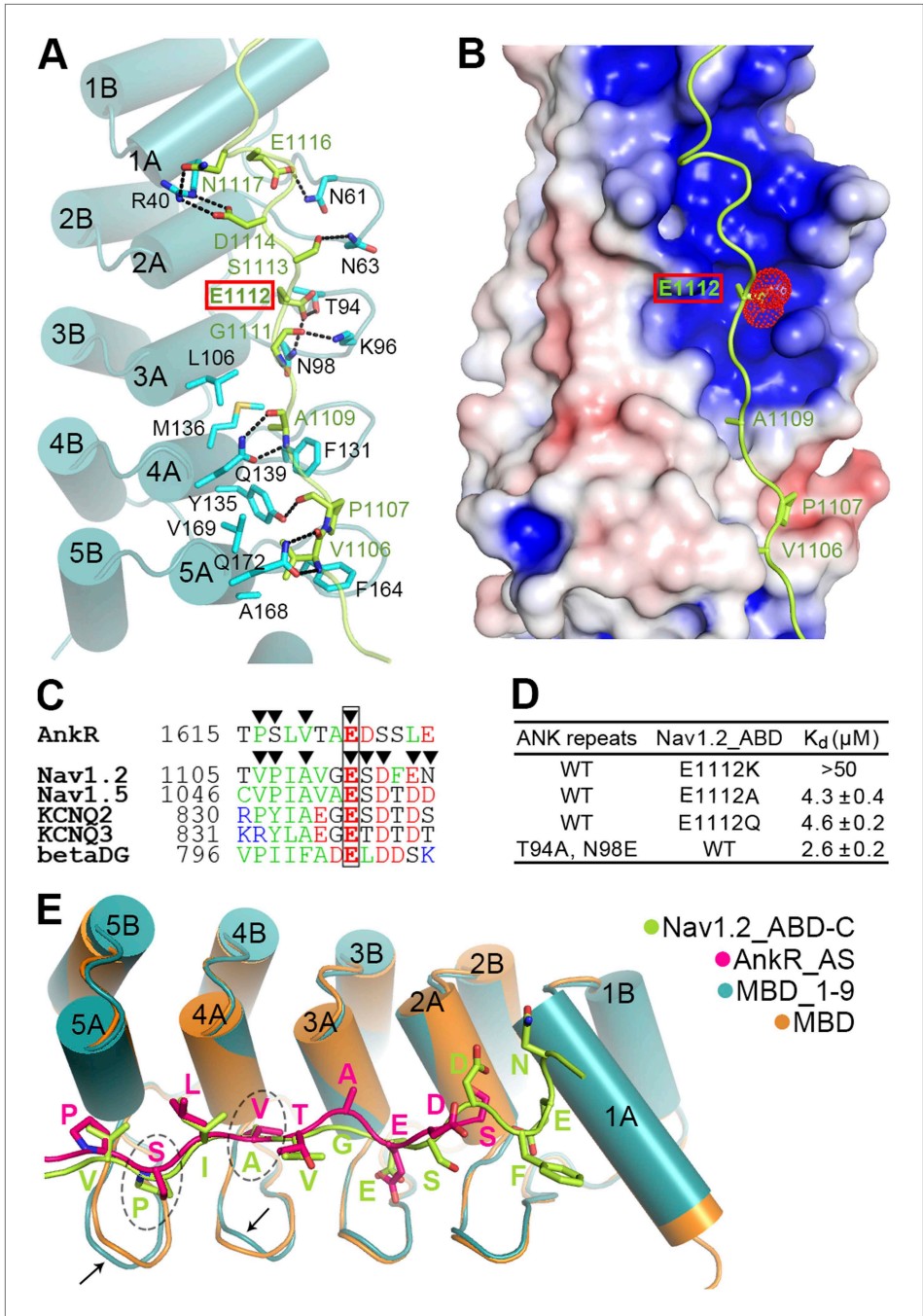

**Figure 6**. Site 1 of ANK repeats is a common binding site for Nav1.2 and other targets. (**A**) Ribbon representation of the binding of Nav1.2_ABD-C (light green) to site 1 of ANK repeats (cyan). The interface residues are shown in the stick mode. Hydrogen bonds and salt bridges are indicated by dashed lines. The negatively charged Glu1112, critical for interacting with a positively charged surface formed by ANK repeats R2 and R3, is highlighted with a red box. (**B**) Charge potential surface of site 1 on the ANK repeats reveals a positively charged pocket important for anchoring of Glu1112 through charge complementation. The hydrophobic groove and the interacting residues from Nav1.2 are also shown. The surface diagram is drawn with the same orientation as in panel **A**. The electrostatic surface potentials were calculated by the APBS module embedded in PyMOL with the non-linear Poisson–Boltzmann equation and contoured at ±5 kT/e. (**C**) Amino acid sequence alignment of the site 1 binding sequences in various partners showing the similar sequence pattern, with the anchoring Glu boxed. The residues participating in site 1 binding are indicated by triangles. (**D**) Summary of the ITC-derived $K_d$ values showing that Glu1112 is essential for the ANK repeats/Nav1.2 interaction. (**E**) Structural comparison of the ANK repeats site 1 bindings of AnkR_AS
*Figure 6. Continued on next page*

*Figure 6. Continued*

and Nav1.2_ABD-C showing that the two targets bind to ANK repeats with essentially the same mode. Subtle conformational differences in the finger loops R4 and R5 are indicated by arrows.

The following figure supplement is available for figure 6:

**Figure supplement 1**. Crystallographic characterization of the ANK repeats/Nav1.2 structure.

the identical position as Glu1622 of AS does in the ANK repeats/AS complex (*Figure 3A* and *Figure 6A,E*). In contrast to the common expectation of directly interacting with positively charged residue(s), Glu1112 of Nav1.2 is buried deeply in the groove and forms hydrogen bonds with the side-chains of Thr94 and Asn98 in the R2 and R3 finger loop (*Figure 6A*). Charge potential calculation shows that the Glu1112 binding pocket formed by R2 and R3 is highly positive, and thus nicely accommodates the negatively charged carboxyl group of Glu1112 (*Figure 6B*). As expected, the charge reversal mutation of Nav1.2 (E1112K) abolished the channel's binding to ANK repeats. Even mild substitutions (E1112Q- and E1112A-Nav1.2) weakened the binding of Nav1.2 to ANK repeats by ~30-fold (*Figure 6D*). In agreement with our findings, E1112Q- and E1112A-Nav1.2 (or E1100 in Nav1.6) failed to cluster at the AIS of hippocampal neurons (*Fache et al., 2004*; *Gasser et al., 2012*). Conversely, substitutions of Thr94 and Asn98 of ANK repeats with Ala and Glu, respectively, also weakened the ANK repeats/Nav1.2_ABD interaction (*Figure 6D*). The above biochemical and structural data illustrate the importance of the absolutely conserved Glu in various ankyrin binding targets shown in *Figure 5D* and *Figure 6C* in anchoring these binding domains to site 1 of ANK repeats. The structures of ANK repeats in complex with the two different targets shown here also provide a framework for understanding the binding of KCNQ2/3, β-dystroglycan, and potentially other ankyrin partners.

## AnkG-mediated clustering of Nfasc and Nav channels at AIS

We next evaluated the consequences of mutations of AnkG characterized in *Figure 3* on its function in clustering Nav channels and Nfasc at the AIS in cultured hippocampal neurons. It is predicted that the 'FF' mutant in site 1 of AnkG_repeats disrupts its Nav1.2 binding but retains the Nfasc binding (*Figure 3F*). As shown previously (*He et al., 2012*), the defect in both AIS formation and Nav channels/Nfasc clustering at the AIS caused by knockdown of endogenous AnkG could be rescued by co-transfection of the shRNA-resistant, WT 270 kDa AnkG-GFP (*Figure 7*). The 'FF' mutant of 270 kDa AnkG-GFP was concentrated normally at the AIS, but failed to rescue clustering of endogenous Nav at the AIS (*Figure 7A,C,D*), consistent with the significantly weakened binding of the mutant AnkG to Nav1.2 (*Figure 3E,F*). This result confirms that the proper clustering of Nav at the AIS depends on AnkG (*Zhou et al., 1998*; *Garrido et al., 2003*). In contrast, Nfasc clustered properly at the AIS in neurons co-transfected with 'FF'-AnkG (*Figure 7B,E*), which was predicted since the 'FF' mutant had no impact on AnkG's binding to Nfasc. Interestingly, both the 'IL' (site 2) and 'LF' (part of site 3) mutants of AnkG-GFP failed to cluster at the AIS of hippocampal neurons (*Figure 7C* and *Figure 7—figure supplement 1*), suggesting that the L1-family members (Nfasc and/or Nr-CAM) or other potential ANK repeats site 2/3 binding targets may play a role in anchoring AnkG at the AIS. Not surprisingly, neither of these mutants can rescue the clustering defects of Nav or Nfasc caused by the knockdown of endogenous AnkG (*Figure 7D,E* and *Figure 7—figure supplement 1*).

## Discussion

Ankyrins are very ancient scaffold proteins present in their modern form in bilaterian animals with their functions greatly expanded in vertebrate evolution (*Cai and Zhang, 2006*; *Hill et al., 2008*; *Bennett and Lorenzo, 2013*). Gene duplications as well as alternative splicing have generated much functional diversity of ankyrins in various tissues in vertebrates. However, the N-terminal 24 ANK repeats of ankyrins have remained essentially the same for at least 500 million years (*Figure 2B* and *Figure 2—figure supplement 3*). In contrast, the membrane targets for ankyrins have expanded greatly in respond to physiological needs (e.g., fast signaling in neurons and heart muscles in mammals) throughout evolution, and these membrane targets almost invariably bind to the 24 ANK repeats of ankyrins. Intriguingly, among about a dozen ankyrin-binding membrane targets identified to date (see review by *Bennett and Healy, 2009*) and those characterized in this study, the ankyrin-binding sequences of these targets are highly diverse. It has been unclear how the extremely conserved ANK repeats can

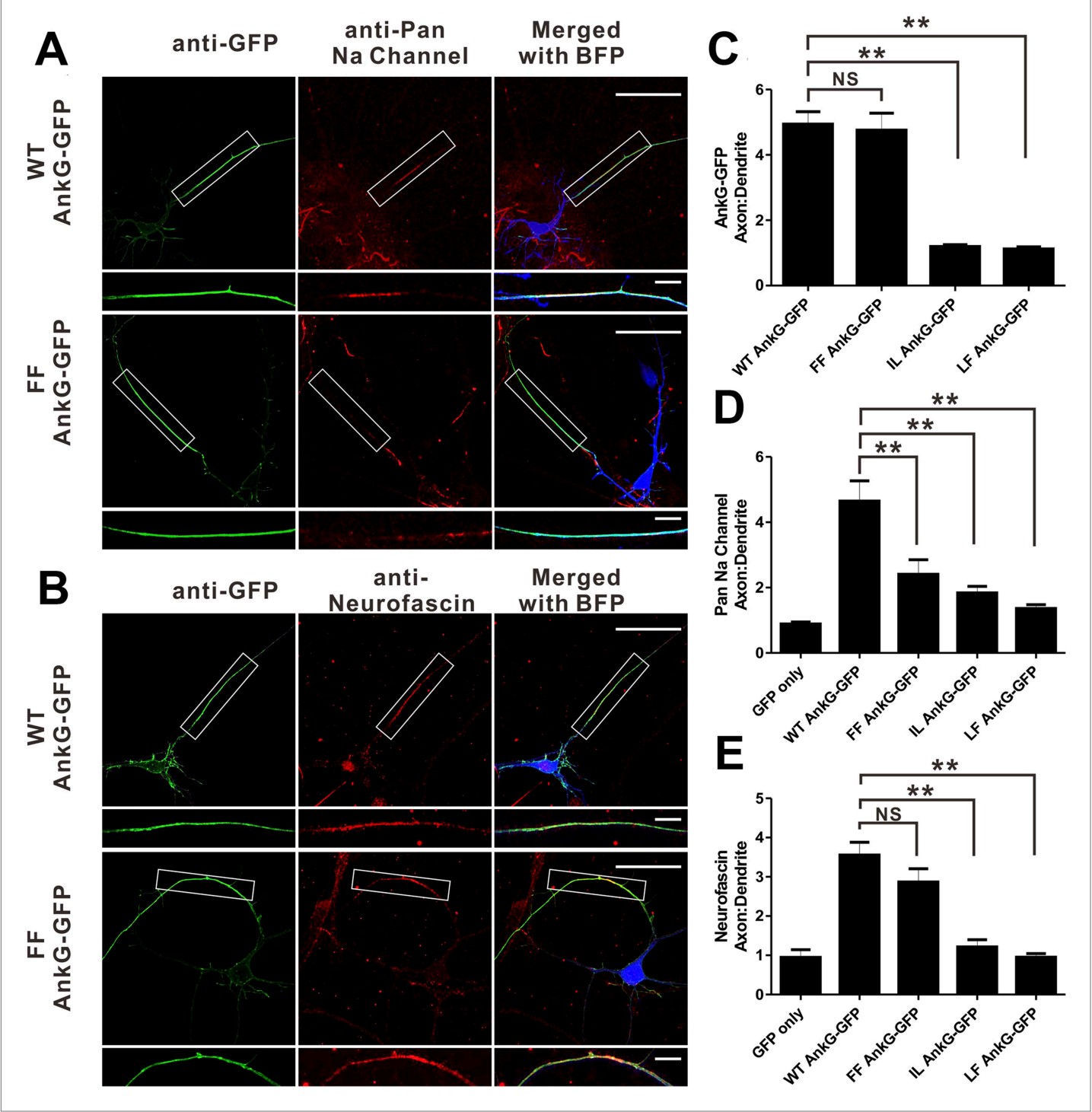

**Figure 7**. Mutations of residues at the target binding groove affect 270 kDa AnkG's function at the AIS in neurons. (**A**) WT 270 kDa AnkG-GFP effectively rescues AnkG self-clustering and clustering of sodium channels at the AIS. The FF mutant of AnkG is clustered at the AIS, but fails to rescue sodium channel clustering at the AIS. BFP marks the shRNA transfected neurons (scale bars, 50 µm). White boxes mark the axon initial segment, which is shown at a higher magnification below each image (scale bars, 10 µm). (**B**) Same as in panel **A** except that the red signals represent anti-neurofascin staining. (**C**) Quantification of anti-GFP fluorescence intensity ratio of axons to dendrites in cells depleted of endogenous 270/480 kDa AnkG and rescued with WT (n = 34), FF (n = 30), IL (n = 24), or LF (n = 24) AnkG-GFP. **p<0.05. Error bars, S.E. (**D**) Quantification of the anti-endogenous pan-sodium channels fluorescence intensity ratio of axons to dendrites in cells depleted of endogenous 270/480 kDa AnkG and rescued with GFP alone (n = 11), WT (n = 17), FF (n = 16), IL (n = 14), and LF (n = 10) AnkG-GFP. **p<0.05. Error bars, S.E. (**E**) Quantification of the anti-endogenous neurofascin fluorescence intensity
*Figure 7. Continued on next page*

*Figure 7. Continued*

ratio of axons to dendrites in cells depleted of endogenous 270/480 kDa AnkG and rescued with GFP alone (n = 6), WT (n = 17), FF (n = 14), IL (n = 10), and LF (n = 10) AnkG-GFP. **p<0.05. Error bars, S.E.

The following figure supplement is available for figure 7:

**Figure supplement 1**. The IL and LF AnkG-GFP mutants do not cluster at the AIS and fail to rescue AnkG's functions in the AIS.

specifically bind to such a diverse set of target sequences. Additionally, it is mechanistically unclear why the membrane targets instead of ANK repeats have undergone amino acid sequence changes in respond to functional diversification in higher vertebrates during evolution. The structure of the entire 24 ANK repeats in complex with an auto-inhibitory domain, together with the structure of part of ANK repeats in complex with its binding domain of Nav1.2, begin to offer insights into the issues above.

## Ankyrin's diverse membrane targets

The 24 ANK repeats form an elongated, continuous solenoid structure with its extremely conserved target binding inner groove spanning a total length of ~210 Å (*Figure 2C*). We identified three distinct target binding sites in the first 14 repeats (*Figure 2* and *Figure 3*). This is in agreement with earlier studies showing that three to five ANK repeats can form a stable structural unit capable of recognizing certain target sequences (*Li et al., 2006*; *Tamaskovic et al., 2012*; *Xu et al., 2012*). Therefore, we predict that the last 10 ANK repeats of ankyrins can contain an additional two to three target binding sites. Importantly, the target binding sites on ANK repeats behave rather independently, as mutations/ disruptions of interactions in each site do not lead to large perturbations in the interactions in the neighboring sites (*Figure 3*). Equal importantly, the ANK repeats targets bind to the inner groove with extended conformations, and the segments responsible for binding to each site do not seem to cooperate with each other (i.e., an alteration in one segment does not have a large impact on the neighboring segments) (*Figure 3* and *Figure 5*). Therefore, the multiple target binding sites on ANK repeats are quasi-independent.

We further show that the AnkR_AS, the Nfasc, the Nav1.2, the KCNQ2, and the Cav1.3 peptides use different combinations of these sites that spread along the elongated and near completely conserved inner ANK repeat groove to form specific ankyrin/target complexes. One can envision that such combinatorial usage of multiple quasi-independent sites can in principle generate a large repertoire of binding targets with different sequences for ANK repeats. Although a number of ion channels use site 1 as the common binding site, the amino acid sequences of the corresponding site 1-binding peptide segments are rather diverse (*Figure 6C*). One can expect that the sequences of target peptide segments responsible for binding to sites 2 and 3 will be even more diverse (e.g., the corresponding site 3 binding sequence of AnkR_AS and Nav1.2 ABD_N have no detectable sequence similarity), as the interactions in these two sites are more hydrophobic in nature (*Figure 3A–C*). The combinatorial usage of the quasi-independent sites, together with the low sequence specificity of each binding site as well as the structural plasticity of the ANK repeat solenoid (*Lee et al., 2006*), indicate that ANK repeats can have large capacities in binding to numerous membrane targets with diverse sequences. In light of the above points, unidentified ANK repeat binding proteins will likely be difficult to predict simply based on amino acid sequences, although a firm conclusion awaits detailed characterizations of more ankyrin binding targets.

The combinatorial usage of multiple binding sites has also been observed in other long repeat-containing proteins including the Karyopherin family nuclear import/export scaffold proteins (*Conti et al., 1998*; *Kobe, 1999*; *Chook and Blobel, 2001*; *Xu et al., 2010*), the Wnt signaling regulatory scaffold β-catenin (*Graham et al., 2000*; *Huber and Weis, 2001*), and tetratricopeptide repeats protein LGN/Pins (*Zhu et al., 2011*). It is possible such a combinatorial target binding strategy may be a common feature for many other elongated repeat-containing proteins in diverse living organisms. The combinatorial multi-site interaction mode may also be advantageous for efficient regulation of ANK repeats/target interactions. By spreading a target binding to multiple sites along the ANK repeats inner groove that are not directly coupled, each binding site can be regulated independently and in a graded fashion. This might allow multiple regulatory signals to be integrated in a combinatorial manner to regulate ankyrin/membrane target interactions. Such a graded regulatory mechanism can be important for ankyrins to respond to various signal inputs when multiple membrane targets co-exist. For example,

AnkG co-exists with Nfasc and sodium and potassium channels at the AIS (*Jenkins and Bennett, 2001*; *Pan et al., 2006*; *Le Bras et al., 2013*), and the components of the AnkG-mediated complex at the AIS can undergo distinct activity-dependent changes (*Hu et al., 2009*; *Grubb and Burrone, 2010*; *Kuba et al., 2010*; reviewed in *Kole and Stuart, 2012)* and exhibit AIS plasticity during development (*Galiano et al., 2012*; *Gutzmann et al., 2014*). It has been reported that Nfasc and sodium channels can undergo activity-dependent phosphorylation in their ANK repeat binding domains (*Garver et al., 1997*; *Whittard et al., 2006*; *Brechet et al., 2008*), which may underlie the distinct patterns of concentration gradients and their activity-dependent changes along the AIS.

## Evolutionary implications of multiple membrane targets of ankyrins

The target binding inner groove of ANK repeats of ankyrins essentially has not changed since the protein evolved over 500 million years ago. In contrast, most, if not all, currently identified ANK repeat-binding segments of ankyrin's membrane targets are either shown or predicted to be unstructured before binding to ankyrins (*Bennett and Lorenzo, 2013*). Such unstructured sequences are more tolerant of mutations as alterations are likely to have a minimal impact on the overall folding of proteins harboring them. Additionally, the conformational malleability is also advantageous for these unstructured peptide sequences to bind to ANK repeats with a molded groove. Since ANK repeats of ankyrins are responsible for binding to numerous targets with diverse sequences, it is likely that there is evolutionary pressure against random mutations in the ANK repeat sequences (the residues in the inner groove in particular). The core function of the ankyrin/spectrin duo in patterning membrane microdomains has remained unchanged throughout evolution, and thus the amino acid sequences of ANK repeats and the spectrin-binding domain of ankyrins are highly conserved (*Wang et al., 2012*). New ankyrin binders (e.g., sodium and potassium channels at the AIS of neurons of higher mammals) can evolve due to functional requirements.

In summary, the structure of the 24 ANK repeats of ankyrins not only offers an explanation for the remarkable capacity of AnkR/B/G to bind to numerous membrane targets, but also provides a framework for guiding future studies of physiological functions and numerous pathological conditions directly associated with ankyrins. Since the three isoforms of ankyrins have distinct physiological functions, highly variable sequences outside the extremely conserved ANK repeats and spectrin-binding domain likely play critical roles in determining the cellular functions of each ankyrin isoform. Finally, proteins with extended ANK repeat domains (e.g., TRP channels, elongation factors, protein kinases and phosphatases, and various scaffold proteins) may also interact with diverse partners via combinatorial uses of multiple, quasi-independent binding sites and thus are particularly suited as adaptors for assembling macromolecular complexes with broad cellular functions.

# Materials and methods

## Constructs, protein expression, and purification

The coding sequences of AnkB_repeats (residues 28–873) were PCR amplified using the full-length human 220 kDa AnkB as the template. The coding sequences of the AnkR constructs, including AnkR_repeats (residues 42–853), and the full length AnkR C-terminal domain (residues 1529–1907), were PCR amplified from a mouse muscle cDNA library. The coding sequence of AnkG_repeats (residues 38–855) were PCR amplified using the full-length rat 190 kDa AnkG as the template. The fusion construct of AnkR_AS and AnkB_repeats was made by standard two-step PCR with a coding sequence of thrombin recognition residues 'GSLVPRGS' as the linker. This construct was used to crystallize and determine the complex structure. The same strategy was used in making other fusion constructs, including the Nav1.2_ABD-C/AnkB_repeats_R1–9 fusion construct containing residues 1103–1129 from mouse Nav1.2 and human AnkB residues 28–318 followed by a capping sequence corresponding to the αB of R24 (residues 814–822) and the AnkR_AS/AnkG_repeats fusion construct. For truncation mutations of ANK repeats constructs, the same capping sequence was added to the appropriate region of the C-terminus of each construct for protein stabilization. Mouse Nav1.2 (NP_001092768.1) and mouse neurofascin (CAD65849.1) were used here for studying their interaction with ankyrins. Peptides for mouse KCNQ2 (NP_034741.2, residues 826–845) and mouse Cav1.3 (NP_083257.2, residues 2134–2166) were commercially synthesized. For simplicity, we used human 220 kDa AnkB for the amino acid numbering throughout the manuscript. For the corresponding point mutations made on AnkG_repeats, each residue number should be increased by 10. All point mutations were created

using the Quick Change site-directed mutagenesis kit and confirmed by DNA sequencing. All of these coding sequences were cloned into a home-modified pET32a vector for protein expression. The N-terminal thioredoxin-His$_6$-tagged proteins were expressed in *Escherichia coli* BL21 (DE3) and purified as previously described (*Wang et al., 2012*). The thioredoxin-His$_6$ tag was removed by incubation with HRV 3C protease and separated by size exclusion columns when needed.

## Isothermal titration calorimetry assay

Isothermal titration calorimetry (ITC) measurements were carried out on a VP-ITC MicroCal calorimeter (MicroCal, Northampton, MA) at 25°C. All proteins were dissolved in 50 mM Tris buffer containing 100 mM NaCl, 1 mM EDTA, and 1 mM DTT at pH 7.5. High concentrations (200–300 µM) of each binding partner assayed in this study, including AnkR_AS, different Nav1.2 ABD proteins and mutants, and neurofascin ABD, were loaded into the syringe, with the corresponding ANK repeats proteins of ankyrin-R/B/G (20–30 µM) placed in the cell. Each titration point was obtained by injecting a 10 µl aliquot of syringe protein into various ankyrin protein samples in the cell at a time interval of 120 s to ensure that the titration peak returned to baseline. The titration data were analyzed using the program Origin 7.0 and fitted by the one-site binding model.

## Analytical gel filtration

Analytical gel filtration chromatography was carried out on an AKTA FPLC system (GE Healthcare, Sweden). Proteins were loaded onto a Superose 12 10/300 GL column (GE Healthcare) equilibrated with a buffer containing 50 mM Tris, 100 mM NaCl, 1 mM EDTA, and 1 mM DTT at pH 7.5.

## Fluorescence assay

Fluorescence assays were performed on a PerkinElmer LS-55 fiuorimeter equipped with an automated polarizer at 25°C. In a typical assay, a FITC (Molecular Probes)-labeled peptide (~1 µM) was titrated with each binding partner in a 50 mM Tris pH 8.0 buffer containing 100 mM NaCl, 1 mM DTT, and 1 mM EDTA. The K$_d$ values were obtained by fitting the titration curves with the classical one-site binding model.

## NMR spectroscopy

For the purpose of NMR analysis, AnkB_repeats fused with AnkR_AS was prepared by growing bacteria in M9 minimal medium supplemented with $^{13}CH_3$-Met (CIL, Cambridge, MA). The protein was expressed and purified using the same method as for the native proteins. Two identical NMR samples containing 0.35 mM of the fusion protein in 50 mM Tris buffer (pH 7.0, with 100 mM NaCl, 1 mM DTT, 1 mM EDTA) were prepared, except that one of the samples contained 50 µg/ml of thrombin. The complete cleavage of the fusion protein was assessed by taking a small aliquot of the thrombin-added sample for SDS-PAGE analysis. NMR spectra were acquired at 35°C on a Varian Inova 750 MHz spectrometer equipped with an actively *z*-gradient shielded triple resonance probe.

## Crystallography

Crystallization of the native AnkR_AS/AnkB_repeats complex and its Se-Met derivative, and the Nav1.2_ABD-C/AnkB_repeats_R1–9 complex was performed using the hanging drop vapor diffusion method at 16°C. Crystals of the ANK repeats/AS complex were obtained from the crystallization buffer containing 0.5 M ammonium sulfate, 1.0 M lithium sulfate, and 0.1 M sodium citrate at pH 5.6. Crystals of the Nav1.2_ABD-C/AnkB_repeats_R1–9 complex were harvested in the crystallization condition with 1.8 M ammonium sulfate, 6–8% dioxane, and 0.1 M MES pH 6.5. Before diffraction experiments, crystals were soaked in crystallization solution containing 30% glycerol for cryoprotection. The diffraction data were collected at Shanghai Synchrotron Radiation Facility and processed and scaled using HKL2000 (*Otwinowski and Minor, 1997*) (*Table 1*).

By using the single isomorphous replacement with anomalous scattering method, the Se-Met sites were found and refined, and the initial phase was determined in AutoSHARP (*Vonrhein et al., 2007*). The structure model of ANK repeats was built manually based on the experimental phase and the last 12 ANK repeats of AnkR_repeats (PBD ID: 1N11) (*Michaely et al., 2002*). Since the AS peptide contains three Met residues, the building of the AS structure was guided by the Se-anomalous difference map as the reference (*Figure 2—figure supplement 2*). Each asymmetric unit contains one ANK repeats/AS molecule. The model was refined against the Se-Met dataset of ANK repeats/AS in PHENIX (*Adams et al., 2002*). COOT (*Emsley and Cowtan, 2004*) was used for model rebuilding and

adjustments. In the final stage, an additional TLS refinement was performed in PHENIX. The initial phase of the Nav1.2_ABD-C/AnkB_repeats_R1–9 complex was determined by molecular replacement using the different repeat regions of the ANK repeats structure as the search models. The Nav1.2_ABD-C peptide was further built into the model. The model was refined using the same strategy as that used for ANK repeats/AS. The model qualities were checked by MolProbity (*Davis et al., 2007*). The final refinement statistics are listed in *Table 1*. All structure figures were prepared by PyMOL (http://www.pymol.org/).

## Hippocampal neuronal cultures

The assay was performed as previously described (*He et al., 2012*). The shRNA of AnkG was cloned into a BFP-pll3.7 vector with the sequence GCGTCTCCTATTAGATCTTTC, targeting a serine-rich region shared by both the 270 kDa and 480 kDa isoforms of mouse AnkG but not the rescuing rat AnkG. Hippocampal neurons were obtained from newborn C57bl/6 mice and cultured until day 4 before co-transfection with the shRNA and different forms of rescue vectors containing rat AnkG using Lipofectamine 2000. On day 7, the neurons were fixed and processed for immunostaining. All the antibodies used in the study were the same as those described in the previous study (*He et al., 2012*).

## Microscopy and data analysis

All the images in this study were captured using a Zeiss LSM 780 laser-scanning confocal microscope. The hippocampal neurons were captured using a 40 × 1.4 oil objective with 0.3 μm Z spacing and pinhole setting to 1 Airy unit. Fluorescence intensity analyses were processed using ImageJ software. The intensity ratios in neurons were quantified and analyzed using GraphPad Prism 5. For the statistical analysis, the neuronal data were compared using one-way ANOVA followed by a Tukey post hoc test.

## Acknowledgements

We thank the Shanghai Synchrotron Radiation Facility (SSRF) BL17U for X-ray beam time, M He, P Jenkins and K Walder for helping with cellular assays, and all members of the MZ and VB laboratories for valuable discussions. This work was supported by grants from RGC of Hong Kong to MZ (663811, 663812, 664113, HKUST6/CRF/10, SEG_HKUST06, AoE/M09/12, and T13-607/12R) and the National Key Basic Research Program of China (2014CB910204 to MZ). MZ is a Kerry Holdings Professor in Science and a Senior Fellow of IAS at HKUST.

## Additional information

### Funding

| Funder | Grant reference number | Author |
| --- | --- | --- |
| Research Grants Council, University Grants Committee, Hong Kong | 663811, 663812, 664113, HKUST6/CRF/10, SEG_HKUST06, AoE/M09/12, and T13-607/12R | Mingjie Zhang |
| Howard Hughes Medical Institute | | Vann Bennett |
| Ministry of Science and Technology of the People's Republic of China | National Key Basic Research Program of China 2014CB910204 | Mingjie Zhang |

The funders had no role in study design, data collection and interpretation, or the decision to submit the work for publication.

### Author contributions

CW, ZW, Conception and design, Acquisition of data, Analysis and interpretation of data, Drafting or revising the article, Contributed unpublished essential data or reagents; KC, FY, CY, Acquisition of data, Analysis and interpretation of data, Contributed unpublished essential data or reagents; VB, Analysis and interpretation of data, Drafting or revising the article, Contributed unpublished essential data or reagents; MZ, Conception and design, Analysis and interpretation of data, Drafting or revising the article, Contributed unpublished essential data or reagents

## Author ORCIDs

Keyu Chen, http://orcid.org/0000-0003-0321-0604

## Additional files

### Major dataset

The following datasets were generated:

| Author(s) | Year | Dataset title | Dataset ID and/or URL | Database, license, and accessibility information |
|---|---|---|---|---|
| Wang C, Wei Z, Chen K, Ye F, Yu C, Bennett V, Zhang M | 2014 | ANK repeats/AnkR_AS complex | http://www.pdb.org/pdb/explore/explore.do?structureId=4rlv | Publicly available at RCSB Protein Data Bank. |
| Wang C, Wei Z, Chen K, Ye F, Yu C, Bennett V, Zhang M | 2014 | Nav1.2_ABD-C/ANK_repeats_R1-9 | http://www.pdb.org/pdb/explore/explore.do?structureId=4rly | Publicly available at RCSB Protein Data Bank. |

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
