## [Decision Letter]

Thank you for sending your work entitled “Structural Basis of Diverse Membrane Target Recognitions by Ankyrins” for consideration at *eLife.* Your article has been favorably evaluated by John Kuriyan (Senior editor), Leemor Joshua-Tor (Reviewing editor), and two reviewers, one of whom, Douglas Barrick, has agreed to reveal his identity.

The editors and the reviewers discussed their comments before we reached this decision, and the Reviewing editor has assembled the following comments to help you prepare a revised submission.

This manuscript describes the structural and energetic basis for recognition of target proteins by the protein Ankyrin. Ankyrin is key protein in assembly of specific membrane complexes and in connecting membrane and cytoskeletal components. The authors solve the structure of the full 24-repeat ankyrin array, a challenging target that has been elusive for many years. To do this, the authors attach recognition sequences to one end of the ankyrin domain in cis. For two different target fusions, diffraction quality crystals were obtained, and structures solved. The structures nicely suggest autoinhibition of AnkR, and guides mutagenesis of potential binding sites for peptides from membrane protein targets.

Importantly, these structures identify a novel mode of recognition by the ankyrin array. There are currently many structures of ankyrin repeat proteins bound to targets, and all of them use short beta hairpins at the junctions separating repeats. Targets of this mode are folded proteins. The current mode of interaction involves an extended polypetide that binds to an elongated groove under the beta-hairpin, and appears to represent a mode by which ankyrin domains can recognize intrinsically disordered regions of targets.

Mutagenesis of 3 distinct sites on AnkR and comparison of effects on binding the autoinhibitory peptide and 4 target peptides suggest combinatorial use of the different sites, explaining sequence diversity of the large number of target peptides that are known to bind ankyrins. A crystal structure of a shorter AnkR repeats fragment bound to a Nav1.2 peptide showed how the target peptide binds 1 of the 3 binding sites that they had characterized.

The authors find that different targets respond differently to these sequence substitutions, and propose that there are different binding determinants within what appear to be overlapping sites. This proposition has interesting implications for how a single surface can develop different specificity for different targets.

However, there are a number of concerns that the author should address:

1) It is unclear how the authors arrived at 3 distinct binding sites for AS. Are the regions between the sites devoid of interactions with the peptide? It is also unclear how the authors concluded that the binding sites are quasi-independent. Independence refers to binding energetics. It doesn't appear that the authors have shown absence of cooperativity between the sites at this time.

At best, the authors might hope their data shows thermodynamic additivity when substitutions from different sites, though this is not shown. And even if it was shown, this does not mean the sites are not coupled, merely that coupling is maintained despite changes in affinity resulting from point substitution and/or fragmentation.

2) Because it was not discussed by the authors, it is unclear if the mode of target peptide binding uncovered here is different from those found in other structures of peptide-bound ankyrin repeat domains such as RFXANK and ANKRA2.

3) Similar combinatorial uses of multiple binding sites to bind diverse peptides have been shown and extensively characterized in other solenoid helical repeat protein systems such as the recognition diverse NLS peptides by the importin Transportin-1 or Karyopherin-beta2.

4) Though there are important new insights from the manuscript, the narrative is hard to follow, and in many cases is overly focused on the details of the mutagenesis and interaction of various targets. It would be help if this myriad of data could be consolidated around a few key ideas and hypotheses. Along these lines, the motivation for switching to fluorescence-based affinity assays from ITC part way through the manuscript is unclear. One of the best motivations would be to measure lower affinity interactions, though this advantage seems not to be used to determine the “ND” interactions from ITC. Note it is unclear that wild-type ANK R-AS fluorescence affinity is well determined. It is quite near the saturation limit, and there appears to be little data defining the saturated baseline.

5) The authors should show electron density for both structures. The resolution of the first structure is rather low, and it would be good to see the crystallographic evidence for specific side-chain interactions (especially charge and hydrogen bond interactions).

6) The methods used to calculate electrostatic potentials should be described.

7) The authors refer to a 1 micromolar binding constant as tight. At best, this as a subjective statement. And to the extent that it is not subjective, it is wrong. Most would regard this as a weak complex.

8) Figure 1: if KDs for AnkR_repeats were measured by ITC, why was the symbol “+++ ” used instead of the actual KD value?

9) Figure 6: it would help if the authors indicate directionality of the peptide or mention that the view is similar to that in Figure 6. It would also be beneficial to enlarge the view in 6C to show the adjacent hydrophobic groove (mentioned in the text) that binds the hydrophobic portion of the peptide.

10) Figure 6: the authors refer to Glu112 participating in charge-charge interactions, but the figure shows only polar interactions to the Glu sidechain. Figure 6 shows that charge complementarity is clearly important. However, use of the term “charge-charge interaction”, when there are no interactions between charged sidechains, is confusing.

11) It would be good if the color scale for conservation in Figure 2 matched. And the average percent identity should be given next to the color scale, rather than “high” and “low”.

12) In the bar graphs in Figure 7, and E, when the authors refer to “LF”, do they mean “LFL”?

---

## [Author Response]

*1) It is unclear how the authors arrived at 3 distinct binding sites for AS. Are the regions between the sites devoid of interactions with the peptide? It is also unclear how the authors concluded that the binding sites are quasi-independent. Independence refers to binding energetics. It doesn't appear that the authors have shown absence of cooperativity between the sites at this time*.

*At best, the authors might hope their data shows thermodynamic additivity when substitutions from different sites, though this is not shown. And even if it was shown, this does not mean the sites are not coupled, merely that coupling is maintained despite changes in affinity resulting from point substitution and/or fragmentation*.

Dividing the AS binding regions in the continuous inner groove of the AnkB_repeats R1-14 into three sites is somewhat arbitrary. One motivation for doing this is that it is easier to convey the key messages derived from our structural and biochemical studies to the broader biological and biomedical research community interested in ankyrin biology. Nonetheless, this division method is supported by or consistent with a large body of experimental data, which would need lots of space to describe and discuss. In order not to distract the main points of the manuscript (please also refer to the point 4 from the reviewers/editors suggesting us to focus on the key points in the manuscript), we decided not to go into such details. The following is a list of some of supporting evidences for such division.

Structurally, there is a fairly clear boundary between each of the two binding sites in the AnkB_repeats/AS complex structure; whereas the interactions within each sites are rather concentrated (Figure 3).

The most direct evidence for dividing AnkB_repeats R1-14 into three sites is probably from the comparison of the large body of the Nav1.2 and Nfsac binding data listed in Figure 3 and Figure 5. In the case of Nav1.2 binding, R1-6 binds to the C-terminal half of the Nav1.2_ABD (ankyrin binding domain) and R11-14 binds to N-terminal half of Nav1.2_ABD. AnkB_repeats R7-10 is not involved in the Nav1.2 binding. In addition, isolated R1-6 is stable and binds to the C-terminal half of the Nav1.2_ABD with a reasonable affinity (Kd ∼2.3 µM; Figure 5), supporting that R1-6 can function as a structural unit for Nav1.2 binding. Thus, one can naturally divide AnkB_repeats R1-14 into three parts, as the middle part is not used for binding. Such division is further supported by the binding data of Nfsac to AnkB_repeats R1-14. Neither the mutations in nor the truncation of R1-6 had any impact on AnkB_repeats R1-14’s binding to Nfasc. Isolated R1-6 has no detectable binding to Nfasc. Thus, Nfasc binds to the R7-14 of AnkB_repeats, and the site-1 is not operational for this binding.

Our division of three binding sites in R1-14 is also based on the well accepted concept that 4-5 ANK repeats can form a folded structural unit. In our case, the sites-2&3 contain 4 ANK repeats each, and the site-1 contains 5 repeats if we do not count the capping repeat 1.

Our experimental data also provide evidence that the three binding sites are largely independent from each other. Since all 24 repeats of ankyrins are folded together as an integral structural unit and their binding targets are all continuous in their amino acid sequences, the interactions among the three sites cannot be totally independent. This is why we used the term quasi-independent in our description. Such description has also been used in the cases of target recognitions by other long repeat scaffold proteins such as Karyopherins and β-catenin (covered in the “Discussion”). The following are a brief summary of some of our data showing that the three binding sites of AnkB_repeats R1-14 are rather independent from each other.

In the case of Nav1.2 binding, the binding affinity contributed by site-1 is with Kd ∼2.3 µM, and that by site-3 is ∼8.1 µM. The Kd of the Nav1.2 to the entire AnkB_repeats is only ∼0.17 µM (Figure 5). This means that the two sites are energetically rather independent from each other, as the Kd of Nav1.2/ AnkB_repeats would be ∼18 pM (2.3 µM X ∼8.1 µM) if the two sites are completely coupled with each other.

The dissections of the interaction sites of AnkB_repeats with AS and with Nfasc reveal a similar picture as that for the Nav1.2/AnkB_repeats interaction. For example, mutations in the site-1 or site-2 each caused a modest decrease (∼20-fold) of AS binding (Figure 4), suggesting that the two sites are not tightly coupled with each other. Similarly, mutations in site-2 and site-3 led to ∼13- and ∼30-fold decreases of Nfasc binding, respectively (Figure 3).

We did not elaborate the above descriptions to avoid the manuscript being excessively lengthy.

*2) Because it was not discussed by the authors, it is unclear if the mode of target peptide binding uncovered here is different from those found in other structures of peptide-bound ankyrin repeat domains such as RFXANK and ANKRA2*.

The largest difference between our AnkB/AS structure and the reported ANKRA2/HDAC4 and RFXANK/RFX5 structures is that AnkB uses a much longer target binding groove than the other two ankyrin repeats in binding to its targets. Nonetheless, the bindings of AnkB R1-6 (or the site-1) to AnkR_AS and to Nav1.2 bear some similarity to those of ANKRA2 and RFXANK in that these targets all bind to the inner groove of the ankyrin repeats with extended conformations (See Figure 8 below). However, obvious differences exist even in these short ankyrin repeats/target interactions. First, the orientation of peptide is reversed in ANKRA2/HDAC4 and RFXANK/RFX5 (Figure 8). Second, the binding of AnkB to its targets involves both the A helices and the connecting finger loops of the ankyrin repeats and target sequences are rather diverse (Figure 3); whereas the target binding residues in ANKRA2 and RFXANK are concentrated within a cluster of residues at the A helices (1A, 2A, 3A, 4A, and 5A) and accordingly these ankyrin repeats bind to a conserved sequence motif. We have discussed this briefly after describing the complex structure of Nav1.2 and AnkB in the revised manuscript.

In order not to distract from the main theme of the manuscript, we did not include Figure 8 below in the revised manuscript.Author response image 1.Comparison of the bindings of AnkR_repeats R1-6 to Nav1.2 with those of ANKRA2 and RFXANK to HDAC4 and RFX5.

*3) Similar combinatorial uses of multiple binding sites to bind diverse peptides have been shown and extensively characterized in other solenoid helical repeat protein systems such as the recognition diverse NLS peptides by the importin Transportin-1 or Karyopherin-beta2*.

We agree with the reviewers/editors that several solenoid helical repeat proteins have been shown to bind to partners through combinatorial usages of multiple binding sites. We have discussed this concept in the third paragraph of the “Discussion”. We also hypothesized that such combinatorial target binding strategy may be a common feature for many other elongated repeat-containing proteins in diverse living organisms. We have cited two more early structural research articles reporting Karyopherin alpha/target interactions ([14]; Kobe et al, 1999) in the revised manuscript.

*4) Though there are important new insights from the manuscript, the narrative is hard to follow, and in many cases is overly focused on the details of the mutagenesis and interaction of various targets. It would be help if this myriad of data could be consolidated around a few key ideas and hypotheses. Along these lines, the motivation for switching to fluorescence-based affinity assays from ITC part way through the manuscript is unclear. One of the best motivations would be to measure lower affinity interactions, though this advantage seems not to be used to determine the “ND” interactions from ITC. Note it is unclear that wild-type ANK R-AS fluorescence affinity is well determined. It is quite near the saturation limit, and there appears to be little data defining the saturated baseline*.

We thank the reviewers/editors for the suggestion. Indeed, the manuscript contains massive amounts of biochemical data involving Ank_repeats point mutations and truncations as well as a panel of ankyrin’s targets in order to provide adequate supports to the scientific conclusions drawn. We have tried further to balance the depth of description of these data and focus on the main points of the manuscript during the revision. We feel that certain level of description of experimental data is necessary in order to maintain the logic and to make the manuscript easily understandable.

Regarding the binding assay methods, we actually began our quantitative biochemical analyses using the fluorescence-based assay, and performed a large number of binding experiments. In the course of our study, we discovered that the hydrophobic fluorescein isothiocyanate (FITC) attached to the N-termini of peptides can artificially enhance their bindings to ANK_repeats, and this occurs when peptides bind to the center of ANK-repeats inner groove (e.g. the sites-2&3 binding by the Nfasc peptide) or when FITC is too close to the key interacting residues in peptides. Because of such complications, we switched to the ITC-based method due to its label-free nature, and the majority of our binding data shown in the manuscript are based on ITC-derived measurements. Nonetheless, a large body of binding measurements derived from FITC-based assay is still valid, and chose a subset of the data to present in Figure 4. We have control experiments to make sure that the FITC-based assay data shown in Figure 4 are free of FITC-induced artifacts, as the binding affinities of the peptides bind to the WT ANK_repeats measured using the two methods (fluorescence-based and ITC-based) are essentially the same (e.g., the AS peptide binding data shown in Figure 1 and Figure 4).

For the WT AnkR_repeats/AS interaction measured by FITC-based assay, the enlarged insertion in Figure 4 (top right) shows that the titration has reached near saturation and the curve can be fitted with a very small error.

*5) The authors should show electron density for both structures. The resolution of the first structure is rather low, and it would be good to see the crystallographic evidence for specific side-chain interactions (especially charge and hydrogen bond interactions)*.

We thank the reviewers/editors for the suggestion of showing more crystallographic evidences. Although the resolution of the first structure is rather low, the electron densities for the binding sites are quite reliable. This is now shown in the updated version of Figure 2—figure supplement 2 panel B. We have also added a new figure (Figure 6—figure supplement 1) to show the electron density of the AnkB/Nav1.2 complex structure.

6) The methods used to calculate electrostatic potentials should be described.

We have added the method for calculating the electrostatic potential in the revised manuscript (Figure 6 legend). In brief, the electrostatic surface potentials were calculated by the APBS module embedded in PyMol with the nonlinear Poisson–Boltzmann equation and contoured at ± 10 kT/e.

*7) The authors refer to a 1 micromolar binding constant as tight. At best, this as a subjective statement. And to the extent that it is not subjective, it is wrong. Most would regard this as a weak complex*.

We agree with the reviewers/editors on this point, and we have made corresponding changes in the revised manuscript.

*8)*
Figure 1*: if KDs for AnkR_repeats were measured by ITC, why was the symbol “+++ ” used instead of the actual KD value?*

In our mapping experiment, we tested the binding through analytical gel filtration chromatography-based assay first and the assay gave quite clear results showing that binding to 1529-1825 of AnkR C-terminal led to an obvious elution volume shift of AnkR_repeats, while 1826-1907 had no interaction at all. So we directly went on to the next round of mapping without measuring the binding affinities of the “1529-1825” fragment using ITC. In response to the comment above, we have performed the ITC experiments during the revision and revised the Figure 1 accordingly. The data is consistent with the expectation that the N-terminal part of AnkR_CT contains the complete AnkR_repeats binding domain by having the same Kd with of the 1529-1907 fragment.

*9)*
Figure 6*: it would help if the authors indicate directionality of the peptide or mention that the view is similar to that in*
Figure 6*. It would also be beneficial to enlarge the view in 6C to show the adjacent hydrophobic groove (mentioned in the text) that binds the hydrophobic portion of the peptide*.

Thanks for the suggestion. Corresponding revisions have been made in the revised Figure 6. The hydrophobic groove and the interacting residues from Nav1.2 have been shown in the panel B of the figure. We have re-arranged the figure accordingly (move panel C to beneath the panel A) to accommodate the change made.

*10)*
Figure 6*: the authors refer to Glu112 participating in charge-charge interactions, but the figure shows only polar interactions to the Glu sidechain.*
Figure 6
*shows that charge complementarity is clearly important. However, use of the term “charge-charge interaction”, when there are no interactions between charged sidechains, is confusing*.

We thank the reviewers/editors for pointing this out to us. We have discussed the importance of the charge potential surfaces of the ANK repeats and the charged nature of Glu1112, as we have demonstrated that this charge potential complementarity is critical for their binding. Although it is not directly charge-charge sidechains interaction, it is electrostatic interaction in nature. To avoid potential confusion, we have modified the description as “The negatively charged Glu1112, critical for interacting with a positively charged surface formed by ANK repeats R2 and R3, is highlighted with a red box” in the revised manuscript.

*11) It would be good if the color scale for conservation in*
Figure 2
*matched. And the average percent identity should be given next to the color scale, rather than “high” and “low”*.

The Figure 2 has been revised following the suggestion. Instead of using the percentage of identity, we calculated the conservation score using the Scorecons server (http://www.ebi.ac.uk/thornton-srv/databases/cgi-bin/valdar/scorecons_server.pl).

*12) In the bar graphs in*
Figure 7*, and E, when the authors refer to “LF”, do they mean “LFL”?*

In the neuronal culture experiments, we actually used a milder mutant for the third binding site, in which only two of the hydrophobic residues in repeats 11 and 12 were changed to Gln (LF to Q). For all the biochemical assays, we chose the “LFL” mutant as the mutation produced more obvious impacts (i.e. at similar levels to those of the site-1 or-2 mutants) in the target bindings via the site-3 (Figure 3 and Figure 4). The “LF” mutant has a milder impact on AnkB’s binding to AS or Nav1.2. In our neuronal culture experiment, we discovered that the milder “LF” mutant in site-3 is sufficient to disrupt AnkG’s AIS clustering. We thus used this milder “LF” mutant instead of the “LFL” mutant in all subsequent multiple batches of neuronal culture experiments. We have specifically mentioned in the main text that the site-3 mutant used in neuronal culture experiments is the “LF” mutant.